

# Impact of foliar application using amino acids, yeast extract, and algae extract in different concentrations on growth parameters, yield traits, grain quality, and nitrogen-related parameters of wheat in arid environments

Mohamed Ebaid[1], Mohamed E. El-Temsah[2],
Mohamed A. Abd El-Hady[2], Amirah S. Alahmari[3],
Ahmed S. M. El-Kholy[4], Diaa Abd El-Moneim[5] and Ahmed M. Saad[6]

[1] Plant Production Department, Arid Lands Cultivation Research Institute (ALCRI), City of Scientific Research and Technological Applications (SRTA-City), New Borg El-Arab City, Alexandria, Egypt
[2] Agronomy Department, Faculty of Agriculture, Ain Shams University, Cairo, Egypt
[3] Department of Biology, College of Science, Princess Nourah bint Abdulrahman University, Riyadh, Saudi Arabia
[4] Agronomy Department, Faculty of Agriculture, Zagazig University, Zagazig, Egypt
[5] Department of Plant Production (Genetic Branch), Faculty of Environmental Agricultural Sciences, Arish University, El-Arish, Egypt
[6] Agronomy Department, Faculty of Agriculture, Benha University, Benha, Egypt

Corresponding author
Diaa Abd El-Moneim,
dabdelmoniem@aru.edu.eg

## ABSTRACT

Wheat cultivation in arid environments faces significant challenges, necessitating innovative approaches to enhance productivity under current climate change conditions. Foliar application with bio-stimulants, such as amino acids, yeast extract, and algae extract, offers a sustainable solution to improve wheat growth, yield, and physiological efficiency under these conditions. A field trial was carried out over two winter growing seasons to assess the comparative effects of these bio-stimulants applied at varying concentrations on growth parameters, yield traits, grain quality, and nitrogen-related parameters. Treatments included an untreated control (sprayed with distilled water) and foliar applications at different concentrations: amino acids (1.5 and 3 ml/L), yeast extract (50 and 100 ml/L), and algae extract (5 and 10 ml/L). The results demonstrated that all bio-stimulant treatments significantly enhanced all studied parameters compared to the untreated control. The highest concentrations of each treatment (3 ml/L for amino acids, 10 ml/L for algae extract, and 100 ml/L for yeast extract) produced the greatest improvements, with amino acids at 3 ml/L exhibiting the strongest effects. Specifically, amino acids at 3 ml/L improved plant height by 12.46% and 21.56%, chlorophyll content by 22.32% and 19.12%, and plant dry weight by 63.30% and 70.38% in the first and the second seasons, respectively. Yield traits, including number of spikes, spike length, spike weight, number of spikelets per spike, number of grains per spike, and 1,000-grain weight, were also significantly improved, with amino acids at 3 ml/L producing the highest values. Grain yield, straw yield, and biological yield increased by 44.74% and 43.92%, 35.34%
and 42.37%, and 36.29% and 43.95%, respectively, in the first and second seasons. Nitrogen content in both grains and straw was higher in treated plants, with amino acids at 3 ml/L enhancing grain nitrogen content by 25.52% and 22.50% and straw nitrogen content by 41.03% and 56.45% in the first and second seasons. The improvement resulted from amino acids at 3 ml/L followed by the application of algae extract at 10 ml/L and yeast extract at 100 ml/L, with all treatments showing significant improvements over the untreated control. Principal component analysis and heatmap analyses confirmed that higher concentrations of amino acids, algae extract, and yeast extract had the most positive effects on wheat growth and nitrogen-related parameters, while the untreated control and lower concentrations showed weaker results. These findings demonstrated that foliar application of amino acids, particularly at 3 ml/L, is a highly effective strategy for improving wheat productivity, grain quality, and nitrogen use efficiency in arid environments.

## INTRODUCTION

Wheat (*Triticum aestivum* L.) is one of the most significant cereal crops in the world and a key component of global food security (*King et al., 2024*). It has a prominent position among food grain crops due to its adaptability to diverse climatic conditions, high nutritional value, and widespread cultivation across various agroecological zones (*Grote et al., 2021*). The increasing global population and shifting dietary preferences have led to growing demand for wheat, necessitating continuous improvements in its production to ensure food availability and stability (*Hossain et al., 2021*). However, achieving sustainable wheat production is challenged by several factors, including soil fertility, climate change, biotic and abiotic stresses, and the depletion of natural resources (*Mansour et al., 2017*; *Rebouh et al., 2023*). Among these factors, soil fertility remains a key determinant of wheat productivity, as it directly influences plant growth, yield potential, and grain quality (*Shah & Wu, 2019*; *Swailam et al., 2021*). Traditional agronomic practices such as fertilization and crop rotation have been widely employed to maintain soil health and enhance wheat production (*Li et al., 2018*). However, excessive use of chemical fertilizers can result in pollution, nutritional imbalances, and soil damage (*Alnass et al., 2023*; *Boudiar et al., 2022*). Therefore, the integration of biologically active substances, including plant growth regulators, organic amendments, and microbial inoculants, has become a viable strategy for improving soil fertility and enhancing wheat productivity in a sustainable manner (*Abd El-hady et al., 2022*; *Elnahal et al., 2022*; *Selem et al., 2022*).

In recent years, there has been a growing interest in applying foliar bio-stimulants, such as amino acids, yeast extracts, and algae extracts, to enhance cereal crop performance across diverse environmental conditions. Several studies reported that foliar bio-stimulants can improve growth, yield, and physiological parameters in cereals by promoting nutrient

uptake, stimulating metabolic activity, and enhancing stress tolerance (*Abd-El-Aty et al., 2024*; *Mansour et al., 2023*). Amino acids play a fundamental role in plant metabolism, serving as the building blocks of proteins and precursors for several bioactive compounds, including vitamins, nucleotides, and plant hormones (*Shiade et al., 2024*). In addition, they are crucial for enzymatic activities, stress tolerance, and overall plant growth and development (*Khan et al., 2020*). Amino acids are significant as a precursor of indole-3-acetic acid (IAA), a key plant growth hormone involved in cell elongation, differentiation, and division (*Zhang et al., 2022*). By encouraging the prolonged release of IAA and enhancing physiological functions like osmotic regulation, ion transport, and stomatal function, amino acid administration can improve plant growth and development (*Al-Turki et al., 2023*). Additionally, amino acids contribute to nitrogen assimilation and protein synthesis, which are critical for the development of biomass and grain formation in wheat (*El-Sanatawy et al., 2021*).

In addition to amino acids, yeast extract is another promising bio-stimulant that boosts plant growth and productivity (*Rajesaheb et al., 2025*). It is a rich natural source of growth-promoting substances, including cytokinins, vitamins (B1, B2, B3, B6, and B12), proteins, carbohydrates, nucleic acids, and essential lipids (*Elsheikh & Eltanahy, 2024*). Yeast extract has been recognized as a sustainable and eco-friendly alternative to chemical fertilizers, offering benefits for both crop production and environmental safety (*Hernández-Fernández et al., 2021*). Yeast extract application could improve root growth, enhance nutrient uptake, and boost crop yield and quality (*Mukherjee et al., 2020*). Moreover, its role in promoting microbial activity in the rhizosphere can further contribute to plant growth and productivity (*Devi et al., 2020*). Besides foliar spray of yeast extracts, it stimulates chlorophyll synthesis, increases photosynthetic efficiency, and enhances tolerance to abiotic stresses such as salinity and drought factors, which are particularly critical in arid environments where wheat cultivation faces significant challenges (*Abdelaal et al., 2021*). Algae extract, whether applied as a foliar spray or added to soil, has also gained attention as an effective bio-stimulant due to its rich composition of essential plant growth regulators (*Han et al., 2024)*. Natural hormones in algae extracts, such as gibberellins, cytokinins, and auxins, promote metabolic and physiological processes in plants, resulting in increased yield and growth (*Tan et al., 2021*). Additionally, algae are rich in polysaccharides, amino acids, and trace elements such as iron, zinc, and manganese, which play a vital role in plant nutrition and stress resistance (*Abideen et al., 2022*). Furthermore, algae-based bio-stimulants have demonstrated their ability to improve soil structure, enhance water retention, and increase microbial diversity, making them highly beneficial for sustainable agriculture (*Kholssi et al., 2022*).

Most previous research focused on single bio-stimulant types or limited concentration ranges, often under non-arid conditions, and comprehensive comparative studies evaluating multiple bio-stimulants and their concentrations in arid environments remain scarce. Hence, the present study aimed to evaluate the effects of foliar applications of amino acids, yeast and algae extracts at varying concentrations on wheat yield, yield components, total chlorophyll content, grain quality, and nitrogen-related physiological parameters under arid conditions. This research seeks to provide insights into the

effectiveness of these natural growth enhancers in optimizing wheat production in semi-arid and arid environments. Understanding the interactive effects of these bio-stimulants in different concentrations on wheat performance could provide insightful information on sustainable farming methods to improve wheat productivity and preserve soil health in arid regions.

## MATERIALS AND METHODS

### Experimental site and applied treatments

Two field experiments were carried out for two winter growing seasons of 2022/2023 and 2023/2024 at the Experimental Research Station, Benha University, Kaloubia Governorate, Egypt (30.35°N, 31.21°E). The properties of the experimental soil are presented in Table 1. The soil is classified as clay, with a textural composition of 24.07% sand, 29.59% silt, and 46.34% clay. Additionally, the site is characterized by an arid and hot climate, marked by high temperatures and low annual rainfall, typical of regions with limited water availability (Table 2).

The applied bio-stimulants in this study were amino acids, yeast extract, and algae extract. The amino acid solution was tyrosine-based (purchased from SHOURA Chemicals, Alexandria, Egypt) and contained micronutrients including zinc (4%), iron (8%), manganese (4%), molybdenum (1%), and boron (3%). Also, it provided 24% free amino acids and 0.03% auxins. The algae extract used in this study, sourced from Egyptian Algae Technology (EAT), El-Obour, Egypt, contained a balanced composition of essential macro- and micro-elements. Its macronutrient profile included nitrogen (6.0%), potassium oxide (1.5%), phosphorus pentoxide (1.5%), magnesium oxide (0.25%), calcium oxide (0.4%), and sulfur (0.8%). The extract also provided micronutrients such as iron (1,000 ppm), zinc (1,000 ppm), manganese (500 ppm), copper (100 ppm), boron (100 ppm), and molybdenum (250 ppm). The yeast extract was prepared from commercial baker yeast (*Saccharomyces cerevisiae*) and is characterized by a rich nutrient profile, containing 45% protein, 7% free amino acids, 1.5% nucleic acids, and 0.5% lipids. Additionally, the yeast extract contains minerals such as potassium (1.8%), magnesium (0.22%), calcium (0.28%), and phosphorus (1.25%). It also supplies B-complex vitamins (including B1, B2, B6, and B12), folic acid, and niacin.

A completely randomized block design (CRBD) with three replicates was used for both growing seasons. The treatments were randomly assigned within the blocks. Each experimental unit consisted of ten rows, each 5 m long, with a 20-cm spacing between the rows. The experiment comprised seven foliar application treatments: control treatment where plants were sprayed with distilled water and six treatments involving amino acids (1.5 and 3 ml/L), yeast extract (50 and 100 ml/L), and algae extract (5 and 10 ml/L). Each treatment was applied twice during the growing season, the first application at 40 days after sowing (DAS) and the second 10 days later (50 DAS). To ensure uniform application across all treatments, a standardized spray volume of 450 liters per hectare was used. The wheat variety Misr 1 was sown on November 25th in both seasons.

**Table 1 Monthly weather data during the field trials in 2022–2023 and 2023–2024 seasons.**

| Month | Minimum temperature (°C) | Maximum temperature (°C) | Relative humidity (%) | Total precipitation (mm) |
|---|---|---|---|---|
| **First season** | | | | |
| 11-2022 | 15.60 | 26.26 | 61.04 | 7.08 |
| 12-2022 | 13.02 | 23.90 | 68.09 | 23.51 |
| 01-2023 | 10.51 | 21.08 | 73.86 | 18.51 |
| 02-2023 | 8.98 | 19.38 | 69.66 | 21.16 |
| 03-2023 | 11.79 | 25.18 | 61.15 | 18.71 |
| 04-2023 | 13.73 | 28.81 | 54.85 | 9.70 |
| **Second season** | | | | |
| 11-2023 | 17.28 | 28.68 | 60.84 | 3.81 |
| 12-2023 | 13.50 | 23.61 | 72.50 | 35.66 |
| 01-2024 | 10.22 | 20.59 | 68.13 | 26.64 |
| 02-2024 | 9.99 | 21.49 | 70.86 | 13.89 |
| 03-2024 | 11.68 | 25.38 | 61.54 | 7.76 |
| 04-2024 | 15.32 | 30.92 | 57.83 | 2.89 |

**Table 2 Properties of the experimental soil across 2022–2023 and 2023–2024 seasons.**

| Properties | 1st season | 2nd season |
|---|---|---|
| Find sand (%) | 18.65 | 17.08 |
| Course sand (%) | 6.92 | 5.49 |
| Clay (%) | 45.28 | 47.40 |
| Silt (%) | 29.15 | 30.03 |
| Texture | Clay | Clay |
| pH | 7.80 | 8.00 |
| E.C. (dS m$^{-1}$) | 0.18 | 0.22 |
| Soluble anions and cations (meq L$^{-1}$) | | |
| $HCO_3$ | 1.19 | 1.23 |
| $Cl^-$ | 0.61 | 0.59 |
| $Na^+$ | 0.79 | 0.83 |
| $Ca^{++}$ | 0.90 | 0.70 |
| $Mg^{++}$ | 0.30 | 0.20 |
| $K^+$ | 0.25 | 0.18 |
| Available nutrient (mg kg$^{-1}$) | | |
| P | 12.0 | 7.00 |
| N | 53.0 | 37.0 |
| K | 384 | 309 |

## Data collection

### Growth and yield parameters

At the flowering stage (90 DAS), ten randomly selected plants from each plot were sampled to measure key growth parameters, including plant height (cm), dry weight per plant (g),

and total chlorophyll content in wheat leaves. Chlorophyll content was determined using a chlorophyll meter (SPAD 402). At harvest, a sample of a square meter was randomly collected from each plot to evaluate yield and yield components, including the number of spikelets per spike, spike length (cm), spike weight (g), 1,000-grain weight (g), biological yield (tons/ha), grain yield (tons/ha), and straw yield (tons/ha).

### Nitrogen-related parameters

The total nitrogen content in both grains and straw was determined using the Micro-Kjeldahl method (*Helrich, 1990*). Grain nitrogen uptake (kg/ha) = (grain nitrogen% × grain yield (kg/ha))/100. Straw nitrogen uptake (kg/ha) = (straw nitrogen% × straw yield (kg/ha))/100. Protein % in grains was calculated by multiplying nitrogen percentage by a factor of 5.70. Additionally, nitrogen physiological parameters were calculated following *Moll, Kamprath & Jackson (1982)*.

Nitrogen uptake efficiency (NUpE) = (Total N uptake/Soil available N + Applied N as fertilizer) × 100. Nitrogen use efficiency (NUE, kg grain/kg N)= Grain yield/(Soil available N + Applied N as fertilizer).

## Statistical analysis

The analysis of variance (ANOVA) was performed separately for each growing season. Treatment means were compared using the least significant difference (LSD) test at a 5% probability level to determine significant differences among treatments. Principal component analysis (PCA) and heatmap visualization were generated using FactoMineR and ggplot2 packages in R, respectively.

## RESULTS

### Growth characteristics

The exogenously sprayed amino acids, yeast extract, and algae extract, at both tested concentrations, significantly increased the total chlorophyll content in wheat leaves compared to the untreated control (Table 3). The highest concentrations of each treatment (3 ml/L for amino acids, 10 ml/L for algae extract, and 100 ml/L for yeast extract) had the most pronounced effect. Among the treatments, amino acids at 3 ml/L exhibited the uppermost plant height (117.3 in the first season and 103.3 in the second season), chlorophyll content (46.77 in the first season and 45.27 in the second season), and plant dry weight (18.45 in the first season and 17.60 in the second season). The application of amino acids at 3 ml/L promoted plant height by 12.46% and 21.56% compared to the untreated control in the first and the second seasons, respectively. Similarly, this application improved chlorophyll content by 22.32% and 19.12%, and plant dry weight by 63.30% and 70.38% in the first and the second seasons, respectively, compared to the untreated control. The enhancement was due to amino acids, followed by the application of algae extract at 10 ml/L and yeast extract at 100 ml/L, with all treatments showing significant improvements over the untreated control. The application of algae extract at 10 ml/L enhanced plant height by 8.31% and 12.55%, chlorophyll content by 19.98% and 18.08%, and plant dry weight by 55.51% and 60.79% in the first and the second seasons,

**Table 3 Effect of foliar application of amino acids, yeast extract, and algae extract on plant height, total chlorophyll content, and plant dry weight of wheat across two growing seasons.**

| Treatment | Plant height (cm) | | Chlorophyll content (SPAD reading) | | Dry weight/plant (g) | |
|---|---|---|---|---|---|---|
| | 1st season | 2nd season | 1st season | 2nd season | 1st season | 2nd season |
| Control | 104.33 e | 85.00 c | 38.23 c | 38.00 e | 11.30 f | 10.33 f |
| Amino-1.5 | 109.67 c | 94.33 b | 45.10 ab | 43.73 bc | 14.67 cd | 14.40 c |
| Amino-3 | 117.33 a | 103.33 a | 46.77 a | 45.27 a | 18.45 a | 17.60 a |
| Algea-5 | 107.67 cd | 90.33 bc | 44.73 ab | 42.70 cd | 13.15 de | 13.17 d |
| Algea-10 | 113.00 b | 95.67 ab | 45.87 ab | 44.87 b | 17.57 ab | 16.61 b |
| Yeast-50 | 106.67 de | 89.00 bc | 42.50 b | 41.87 d | 12.96 e | 11.63 e |
| Yeast-100 | 110.00 c | 95.00 b | 45.70 ab | 43.90 bc | 16.00 bc | 15.04 c |
| **ANOVA** | **Mean squares and significance** | | | | | |
| Treatment | 55.65** | 103.19** | 25.58** | 20.93** | 20.41** | 20.35** |

Notes:
Means with different letters are significantly different according to the LSD test ($P < 0.05$).
** $P < 0.01$.

respectively, compared to the untreated control. Likewise, the application of yeast extract at 100 ml/L stimulated plant height by 5.43% and 11.76%, chlorophyll content by 19.53% and 15.53%, and plant dry weight by 41.59% and 45.60% in the first and the second seasons, respectively, compared to the untreated control.

## Yield traits

The foliar application of amino acids, yeast extract, and algae extract at both tested concentrations significantly increased the number of spikes per square meter, spike length, spike weight, number of spikelets per spike, number of grains per spike, and 1,000-grain weight compared to untreated plants (Table 4). Among the treatments, amino acids at 3 ml/L produced the highest values across all measured spike characteristics, with significant differences observed (Table 4). The application of amino acids at 3 ml/L improved the number of spikes by 36.44% and 42.12% compared to the untreated control in the first and the second seasons, respectively. Likewise, this application improved spike length by 45.61% and 48.14%, spike weight by 59.00% and 58.97%, number of spikelets per spike by 43.27% and 33.33%, number of grains per spike by 31.15% and 30.77%, and 1,000-grain weight by 20.32% and 23.52% in both seasons in the same order compared to untreated control. The enhancement was followed by the application of algae extract at 10 ml/L and yeast extract at 100 ml/L, with all treatments showing significant improvements over the untreated control. The application of algae extract at 10 ml/L enhanced the number of spikes by 32.45% and 30.31% in the first and the second seasons, respectively, compared to the untreated control. Also, this application promoted spike length by 38.51% and 40.74%, spike weight by 42.90% and 42.06%, number of spikelets per spike by 34.79% and 25.93%, number of grains per spike by 19.14% and 27.69%, and 1,000-grain weight by 17.92% and 20.81% in the first and the second seasons, respectively compared to the untreated control. Likewise, the application of yeast extract at 100 ml/L

**Table 4 Effect of foliar application of amino acids, yeast extract, and algae extract on number of spikes per square meter, spike length, spike weight, number of spikelets per spike, number of grains per spike, 1,000-grain weight of wheat across two growing seasons.**

| Treatment | No. spikes/m$^2$ | | Spike length (cm) | | Spike weight (g) | | No. spikelets/spike | | No. grains/spike | | 1,000 grains Weight (g) | |
|---|---|---|---|---|---|---|---|---|---|---|---|---|
| | 1st season | 2nd season | 1st season | 2nd season | 1st season | 2nd season | 1st season | 2nd season | 1st season | 2nd season | 1st season | 2nd season |
| Control | 377.0 c | 338.7 e | 9.867 e | 9.000 d | 23.00 e | 22.67 e | 15.33 c | 18.00 d | 46.33 c | 43.33 e | 50.03 f | 47.90 f |
| Amino-1.5 | 458.0 ab | 390.7 cd | 12.867 c | 11.667 bc | 28.53 cd | 29.20 c | 19.77 ab | 22.67 ab | 53.7 b | 49.67 cd | 55.37 d | 54.07 d |
| Amino-3 | 514.4 a | 481.3 a | 14.367 a | 13.333 a | 36.57 a | 36.03 a | 21.97 a | 24.00 a | 60.77 a | 56.67 a | 60.20 a | 59.17 a |
| Algea-5 | 451.5 ab | 370.7 de | 12.500 cd | 11.000 c | 25.80 de | 25.37 d | 19.07 b | 22.00 bc | 47.70 c | 45.67 de | 54.40 e | 52.93 e |
| Algea-10 | 499.4 a | 441.3 ab | 13.667 b | 12.667 ab | 32.87 b | 32.20 b | 20.67 ab | 22.67 ab | 55.20 b | 55.33 ab | 59.00 b | 57.87 b |
| Yeast-50 | 396.7 bc | 348.0 e | 12.400 d | 10.667 c | 23.70 e | 23.50 e | 18.93 b | 20.67 c | 47.10 c | 44.67 e | 53.97 e | 52.40 e |
| Yeast-100 | 483.2 a | 424.0 bc | 13.333 b | 11.667 bc | 30.67 bc | 30.53 bc | 20.03 ab | 22.67 ab | 54.90 b | 52.00 bc | 57.77 c | 56.73 c |
| ANOVA | Mean squares and significance | | | | | | | | | | | |
| Treatment | 7,893** | 8,154** | 6.15** | 5.97** | 74.52** | 71.95** | 12.79** | 11.43** | 86.30** | 83.82** | 36.11** | 44.34** |

Notes:
Means with different letters are significantly different according to the LSD test ($P < 0.05$).
** $P < 0.01$.

enhanced the number of spikes by 28.16% and 25.20% in the first and the second seasons, respectively, compared to the untreated control. In addition, this application improved spike length by 35.13% and 29.63%, spike weight by 33.33% and 34.70%, number of spikelets per spike by 30.65% and 25.93%, number of grains per spike by 18.49% and 20.00%, and 1,000-grain weight by 15.46% and 18.44% in the first and the second seasons, respectively compared to untreated control.

The results in Fig. 1 demonstrated that foliar application of amino acids, yeast extract, and algae extract at both tested concentrations significantly increased grain yield compared to untreated plants. Among the treatments, amino acids at 3 ml/L resulted in the highest yield traits, with significant differences observed (Fig. 1). The application of amino acids at 3 ml/L enhanced grain yield by 44.74% and 43.92%, straw yield by 35.34% and 42.37%, and biological yield by 36.29% and 43.95% compared to the untreated control in the first and the second seasons, respectively. The improvement was followed by the application of algae extract at 10 ml/L and yeast extract at 100 ml/L, with all treatments showing significant improvements over the untreated control. The algae extract application significantly increased grain yield, straw yield, and biological yield, showing a comparable effect at both 50 and 100 ml/L. The application of algae extract at 10 ml/L stimulated grain yield by 32.34% and 25.76%, straw yield by 31.82% and 27.96%, and biological yield by 33.26% and 37.27% compared to the untreated control in the first and the second seasons, respectively. A similar trend was observed with yeast extract, where both tested concentrations improved yield traits with a higher performance of 100 ml/L (Fig. 1). The application of yeast extract at 100 ml/L promoted grain yield by 28.25% and 19.20%, straw yield by 30.19% and 25.99%, and biological yield by 27.20% and 29.63% compared to the untreated control in the first and the second seasons, respectively.

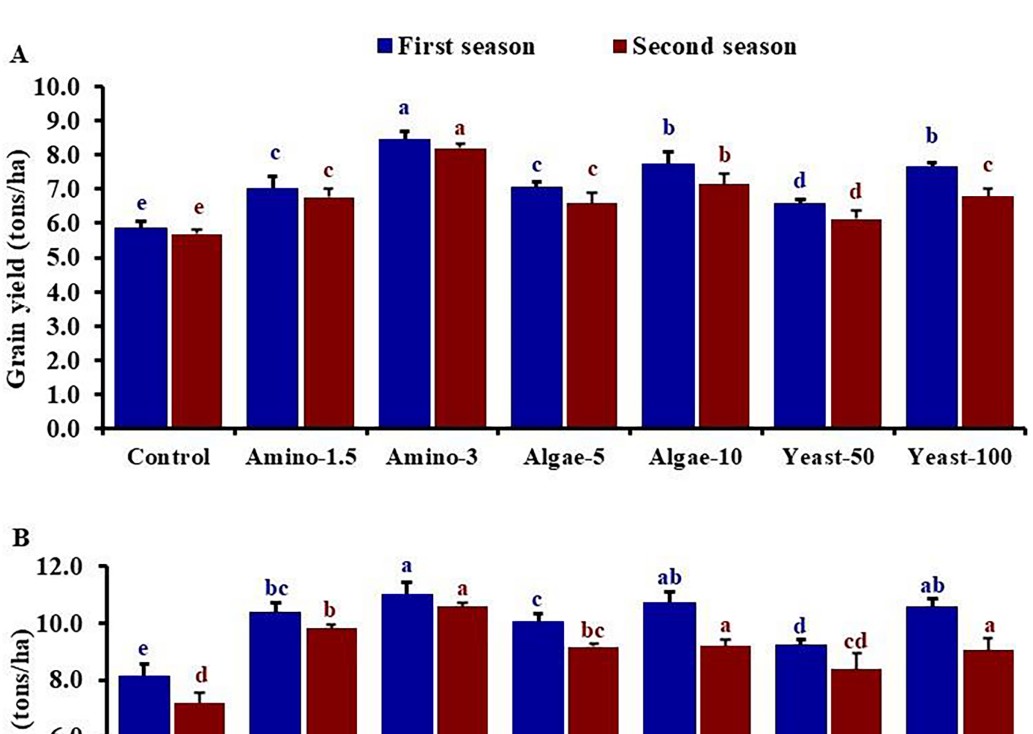

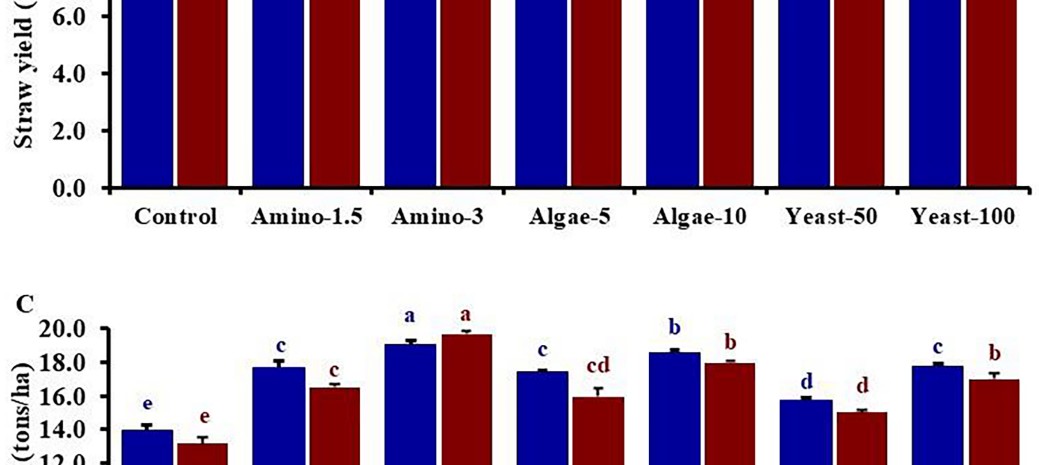

**Figure 1 Effect of foliar application of amino acids, yeast extract, and algae extract on grain yield (A), straw yield (B), and biological yield (C) of wheat across two growing seasons.** The bars on the columns represent the standard deviation and different letters differ significantly by LSD ($P < 0.05$) for each season.

## Nitrogen measurements

The nitrogen content absorbed and accumulated in grains and straw of wheat plants was evaluated across two seasons, comparing the different foliar application treatments. The results presented in Table 5 indicate that the highest nitrogen fractions in both grains and straw were observed in wheat plants treated with amino acids at 3 ml/L. In contrast, wheat plants that were only sprayed with distilled water (control) exhibited the lowest nitrogen content, highlighting the significant impact of natural extract applications on nitrogen assimilation. The application of amino acids at 3 ml/L enhanced grain nitrogen content by 25.52% and 22.50% compared to the untreated control in the first and the second seasons, respectively. Likewise, it improved grain protein content by 25.48% and 22.45% and straw nitrogen content by 41.03% and 56.45% compared to the untreated control in the first and the second seasons, respectively. This improvement was followed by the application of algae extract at 10 ml/L and yeast extract at 100 ml/L, with all treatments showing significant improvements over the untreated control. The application of algae extract at 10 ml/L improved grain nitrogen content by 15.54% and 15.77%, grain protein content by 15.52% and 15.72%, and straw nitrogen content by 29.23% and 39.28% compared to the untreated control in the first and the second seasons, respectively. Similarly, yeast extract at 100 ml/L promoted grain nitrogen content by 6.90% and 13.14% grain protein content by 6.87% and 13.11% and straw nitrogen content by 19.49% and 35.60% compared to the untreated control in the first and the second seasons, respectively. To further assess nitrogen efficiency: nitrogen uptake efficiency (NUpE) and nitrogen use efficiency (NUE) were analyzed (Table 5). The results indicate significant variations in nitrogen efficiency parameters depending on the type and concentration of foliar treatment applied. Among the treatments, amino acids at 3 ml/L exhibited the highest values for NUpE and NUE across both seasons, demonstrating its effectiveness in enhancing nitrogen uptake and utilization by wheat plants. This improvement was followed by the application of algae extract at 10 ml/L and yeast extract at 100 ml/L, with all treatments showing significant improvements over the untreated control. Conversely, the control treatment (distilled water) recorded the highest values, indicating that untreated plants had a lower capacity to absorb and utilize nitrogen effectively. The increased nitrogen efficiency observed in treated plants can be directly linked to their improved nitrogen uptake capacity as shown in Table 5.

## Principal component and heatmap analyses

To evaluate the effects of foliar application treatments on wheat growth, yield, and nitrogen-related parameters, PCA and heatmap visualization were employed (Fig. 2). These multivariate statistical tools allow for the simultaneous assessment of multiple traits and treatments, providing an integrated overview of the data. PCA is a statistical method that reduces the complexity of large datasets by summarizing the variation among all measured traits into a few new variables (principal components). In this study, the first principal component (PC1) accounted for 95.02% of the total variation, while the second principal component (PC2) explained an additional 3.44%. PC1 primarily differentiates treatments based on their overall effectiveness in improving wheat growth, yield, and

**Table 5 Effect of foliar application of amino acids, yeast extract, and algae extract on nitrogen content and efficiencies of wheat across two growing seasons.**

| Treatment | Grain nitrogen content (%) | | Grains protein content (%) | | Straw nitrogen content (%) | | Nitrogen uptake efficiency (kg N absorbed/kg N available) | | Nitrogen use efficiency (kg grain/kg N available) | |
|---|---|---|---|---|---|---|---|---|---|---|
| | 1st season | 2nd season | 1st season | 2nd season | 1st season | 2nd season | 1st season | 2nd season | 1st season | 2nd season |
| Cotrol | 2.366 e | 2.399 e | 13.49 e | 13.68 e | 0.650 e | 0.543 e | 44.98 f | 52.47 f | 15.03 e | 17.27 e |
| Amino-1.5 | 2.474 cd | 2.576 d | 14.10 cd | 14.68 d | 0.727 d | 0.683 e | 64.04 d | 73.23 d | 18.05 c | 20.52 c |
| Amino- 3 | 2.970 a | 2.938 a | 16.93 a | 16.75 a | 0.917 a | 0.850 a | 90.52 a | 105.68 a | 21.75 a | 25.81 a |
| Algea- 5 | 2.470 cd | 2.576 d | 14.08 cd | 14.68 d | 0.707 d | 0.697 cd | 63.00 d | 70.97 d | 18.12 c | 20.013 c |
| Algea- 10 | 2.734 b | 2.777 b | 15.58 b | 15.83 b | 0.840 b | 0.757 b | 76.70 b | 90.24 b | 19.59 b | 23.24 b |
| Yeast- 50 | 2.443 de | 2.419 e | 13.92 de | 13.79 e | 0.687 d | 0.687 d | 57.55 e | 62.44 e | 16.90 d | 18.64 d |
| Yeast- 100 | 2.529 c | 2.714 c | 14.42 c | 15.47 c | 0.777 c | 0.737 bc | 71.39 c | 80.58 c | 19.87 b | 20.59 c |
| ANOVA | Mean squares and significance | | | | | | | | | |
| Treatment | 0.133** | 0.114** | 4.31** | 3.69** | 0.064** | 0.047** | 632.8** | 936.8** | 14.29** | 24.42** |

Notes:
Means with different letters are significantly different according to the LSD test ($P < 0.05$).
** $P < 0.01$.

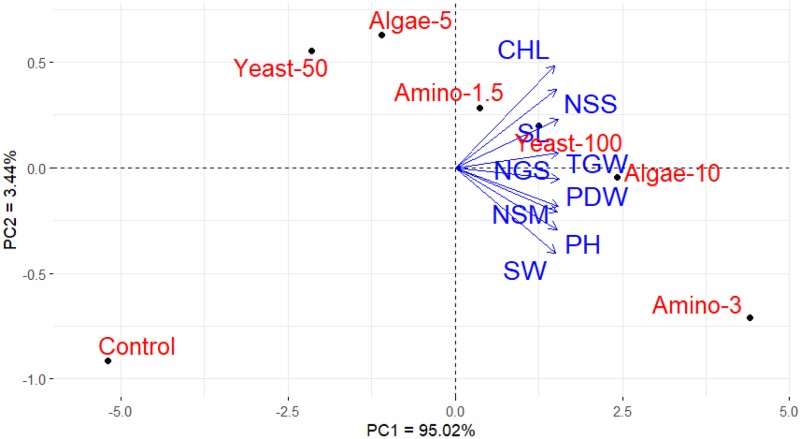

**Figure 2 PCA biplot illustrating the relationship between foliar nutrition treatments and wheat growth and yield parameters.** The applied treatments were untreated control (distilled water), amino acids at concentrations of 1.5 ml/L (Amino-1.5) and 3 ml/L (Amino-3), yeast extract at concentrations of 50 ml/L (Yeast-50) and 100 ml/L (Yeast-100), and algae extract at concentrations of 5 ml/L (Algae-5) and 10 ml/L (Algae-10).

nitrogen-related parameters. Treatments with higher concentrations of amino acids (3 ml/L), algae extract (10 ml/L), and yeast extract (100 ml/L) are positioned toward the positive end of PC1, indicating their strong positive impact on the measured traits. In contrast, the untreated control and lower concentration treatments are located toward the negative end of PC1, reflecting their weaker effects. The PCA plot also shows that the agronomic parameters (such as plant height, yield, and nitrogen uptake) are closely associated with the higher concentration treatments, as indicated by the direction and length of the arrows, which represent the strength and direction of these associations.

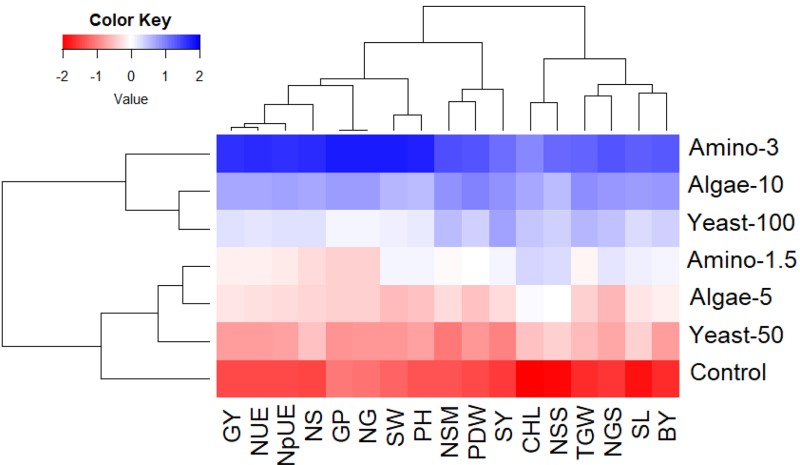

**Figure 3 Heatmap showing the impact of foliar nutrition treatments on growth, yield, and nitrogen-related parameters of wheat in arid environments.** The applied treatments were untreated control (distilled water), amino acids at concentrations of 1.5 ml/L (Amino-1.5) and 3 ml/L (Amino-3), yeast extract at concentrations of 50 ml/L (Yeast-50) and 100 ml/L (Yeast-100), and algae extract at concentrations of 5 ml/L (Algae-5) and 10 ml/L (Algae-10). The assessed parameters were grain yield (GY), nitrogen use efficiency (NUE), nitrogen uptake efficiency (NpUE), straw nitrogen content (NS), grains protein content (GP), grains nitrogen content (NG), spike weight (SW), plant height (PH), Number of spikes/m 2 (NSM), plant dry weight (PDW), straw yield (SY), chlorophyll content (CHL), number of spikelets per spike (NSS), 1,000-grain weight (TGW), number of grains per spike (NGS), spike length (SL), and biological yield (BY).

The heatmap provides a visual summary of how each treatment influences the various measured parameters. In this visualization, each cell represents the value of a specific trait under a particular treatment, with the color indicating the magnitude of the effect. The color scale ranges from red (lower values, indicating poorer performance) through white (intermediate values) to blue (higher values, indicating better performance). Thus, blue regions correspond to treatments and traits with the most favorable outcomes, while red regions indicate weaker or baseline performance. The treatments with higher concentrations of amino acids (3 ml/L), algae extract (10 ml/L), and yeast extract (100 ml/L) are represented by blue regions across most traits, highlighting their superior effects on wheat growth, yield, and nitrogen efficiency (Fig. 3). Conversely, the control treatment is predominantly red, reflecting its lower performance across the same parameters. The clustering patterns in the heatmap further demonstrate that higher concentration treatments group together, emphasizing their similar and pronounced positive impacts.

## DISCUSSION

Arid environments are characterized by high temperatures, low soil fertility, and limited moisture, all of which negatively impact wheat development, grain yield, and quality. These challenging conditions necessitate innovative and sustainable approaches to enhance wheat growth and productivity (*Desoky et al., 2021*). Foliar applications with

bio-stimulants represent a promising strategy, particularly for arid regions where conventional methods are often less effective.

The present study evaluated the comparative effects of amino acids (1.5 and 3 ml/L), yeast extract (50 and 100 ml/L), and algae extract (5 and 10 ml/L) on wheat growth parameters, yield traits, grain quality, and nitrogen-related parameters under arid conditions. The results demonstrated that foliar application of all three bio-stimulants significantly improved key growth characteristics, including chlorophyll content, plant height, and plant dry weight, compared to the untreated control. Notably, the greatest improvements were observed at the highest application rates: 3 ml/L for amino acids, 10 ml/L for algae extract, and 100 ml/L for yeast extract.

The observed enhancements in growth can be attributed to the positive effects of these natural bio-stimulants on plant physiological functions. Increased chlorophyll content in treated plants indicates improved photosynthetic efficiency, which directly contributes to higher biomass production and grain yield. Amino acids, yeast, and algae extracts are rich in essential compounds such as indole-3-acetic acid (IAA), cytokinins, gibberellins, vitamins, and micronutrients, all of which play crucial roles in chlorophyll biosynthesis and photosynthetic activity. In this context, *Sadak, Abdelhamid & Schmidhalter (2015)*, *Khan et al. (2019)* and *Sadak et al. (2023)* elucidated the positive impact of amino acids on plant growth by stimulating IAA production, leading to greater chlorophyll formation.

Yeast extract, in particular, acts as a growth-promoting substance by contributing to stress tolerance and prolonging chlorophyll retention, thereby ensuring higher photosynthetic efficiency under adverse environmental conditions (*Abd El-Sattar & Abdelhameed, 2024*; *Abdelaal et al., 2021*; *Taha et al., 2020*). The significant increases in plant height observed in treated wheat plants are likely due to the hormonal influence of these bio-stimulants, especially through IAA-mediated cell elongation. Amino acids serve as precursors for plant hormones, enhancing nutrient translocation, protein synthesis, and enzymatic activity, which collectively promote internodal elongation and stem growth (*Guo et al., 2021*). Similarly, both yeast and algae extracts promote cell division, elongation, and structural integrity, leading to increased plant height (*Parmar et al., 2023*). In addition, foliar application of amino acids, yeast extract, and algae extract significantly promoted plant dry weight. This increase can be attributed to improved nutrient uptake, enhanced protein synthesis, and greater enzymatic activity in treated plants. Amino acids function as metabolic precursors for growth hormones, leading to higher biomass accumulation (*Salinas et al., 2019*). Similarly, yeast and algae extracts contribute to nutritional balance and stress resilience, further enhancing vegetative growth (*Carillo et al., 2020*; *Gholami et al., 2023*).

A significant improvement in spike length, spike weight, number of spikes per plant, number of spikelets per spike, number of grains per spike, and 1,000-grain weight was observed in treated plants. Amino acids contribute positively to spike differentiation and grain formation (*Ahmad et al., 2019*). Yeast and algae extract further enhanced reproductive growth by increasing nutrient availability, enzyme activity, and growth hormone synthesis, thereby boosting grain setting and spike formation (*Alharbi et al., 2022*; *Hammad & Ali, 2014*). The application of amino acids at 3 ml/L resulted in the

highest grain yield, followed by algae extract at 10 ml/L and yeast extract at 100 ml/L. The increase in grain yield can be attributed to promoted nitrogen assimilation and increased nitrogen (N), phosphorous (P), and potassium (K) uptake in treated plants, improved grain filling, and kernel development. Stimulation of hormonal activity, as the foliar treatments promoted IAA, gibberellin, and cytokinins synthesis, leading to greater carbohydrate partitioning and grain filling. Improved physiological efficiency and higher photosynthetic rates, chlorophyll retention, and stress tolerance resulted in superior grain productivity (*Tshikunde et al., 2019*). The highest straw yield was recorded with amino acids at 3 ml/L, with significant increases observed for yeast and algae extract treatments. The observed improvement in biological yield is likely due to improved microbial activity, improved enzymatic function, and greater nutrient solubility in the soil. Additionally, yeast extract at 50 and 100 ml/L significantly increased straw yield. The highest nitrogen accumulation was recorded in wheat plants treated with amino acids at 3 ml/L. These findings suggest that amino acids enhance nitrogen absorption, assimilation, and translocation from vegetative tissues to reproductive organs. Similarly, yeast and algae extract significantly improved nitrogen content in wheat plants. Amino acids at 3 ml/L recorded the highest nitrogen uptake and use efficiencies, indicating superior nitrogen uptake and utilization. These findings suggest that foliar treatments optimize nitrogen efficiency, reduce losses, and improve overall plant performance. The application of bio-stimulants not only improves plant metabolism but also enhances nutrient use efficiency, reducing the dependence on synthetic fertilizers and minimizing their environmental effects.

Principal component analysis and heatmap analyses played a crucial role in visualizing the relationships between the different foliar treatments and their effects on wheat growth and yield traits (*ElShamey et al., 2022*; *Omar et al., 2022*; *Salem et al., 2020*). The PCA provided valuable insight into the contribution of evaluated treatments and their associated parameters; also, the heatmap analysis helped to clearly illustrate the patterns of improvement in growth, yield, and nitrogen-related traits, emphasizing the effectiveness of applied treatments of amino acids, algae extract, and yeast extract in enhancing wheat performance. Both analyses demonstrated that higher concentrations of amino acids, algae extract, and yeast extract were associated with better wheat performance, while the untreated control and lower concentrations of the treatments exhibited weaker effects. These findings suggest that the concentration of foliar treatments plays a crucial role in determining their effectiveness, with higher concentrations yielding more pronounced improvements in growth, yield, and nitrogen utilization. Hence, these treatments have the potential to be integrated into wheat production systems in arid environments to improve productivity, nitrogen efficiency, and overall plant health.

The economic feasibility of foliar treatments is a critical consideration for their practical adoption by farmers, particularly in arid environments where resource efficiency is essential. The findings of this study indicated that foliar application of amino acids at 3 ml/L significantly improved wheat growth, yield, and nitrogen-use efficiency, hence

representing a cost-effective strategy for improving wheat productivity in particular in arid environments. By improving nitrogen uptake and utilization, amino acid foliar sprays can lower the required fertilizer input, thereby reducing production costs and minimizing environmental pollution associated with excessive fertilizer use. Moreover, the demonstrated yield enhancement suggests a favorable return, making this approach economically applicable and sustainable. This indicates the benefits of foliar amino acid treatments in enhancing crop performance and supporting more sustainable nutrient management practices. The foliar application of amino acid is relatively low-cost and can be integrated with existing farm management practices. Consequently, foliar application of amino acid at 3 ml/L offers a promising, practical solution to increase wheat productivity and nitrogen efficiency, contributing to improved food security and environmental sustainability in arid regions.

## CONCLUSIONS

This present study demonstrated that foliar application of amino acids, yeast extract, and algae extract significantly improved wheat growth, yield, and nitrogen-related parameters under arid conditions. Among the treatments, amino acids at 3 ml/L exhibited the greatest enhancement in plant height, chlorophyll content, dry weight, and yield components such as spike number, spike weight, and grain count per spike. Foliar application also improved nitrogen content, uptake efficiency, and use efficiency, with amino acids outperforming the other extracts. While algae extract (10 ml/L) and yeast extract (100 ml/L) also contributed to improved wheat productivity, their effects were slightly less pronounced. Multivariate analyses, including principal component and heatmap assessments, confirmed the strong association between higher treatment concentrations and better wheat performance. The obtained results indicated that foliar application of these natural extracts, particularly amino acids, is an effective strategy to enhance wheat growth and nitrogen efficiency in nutrient-limited, arid environments.

### Funding

This research was funded by Princess Nourah bint Abdulrahman University Researchers Supporting Project number (PNURSP2025R461), Princess Nourah bint Abdulrahman University, Riyadh, Saudi Arabia. The funders had no role in study design, data collection and analysis, decision to publish, or preparation of the manuscript.

### Grant Disclosures

The following grant information was disclosed by the authors:
Princess Nourah bint Abdulrahman University: PNURSP2025R461.

### Competing Interests

Diaa Abd El Moneim is an Academic Editor for PeerJ.

## Author Contributions

- Mohamed Ebaid conceived and designed the experiments, analyzed the data, prepared figures and/or tables, authored or reviewed drafts of the article, and approved the final draft.
- Mohamed E. El-Temsah performed the experiments, analyzed the data, prepared figures and/or tables, authored or reviewed drafts of the article, and approved the final draft.
- Mohamed A. Abd El-Hady conceived and designed the experiments, authored or reviewed drafts of the article, and approved the final draft.
- Amirah S. Alahmari conceived and designed the experiments, performed the experiments, prepared figures and/or tables, authored or reviewed drafts of the article, and approved the final draft.
- Ahmed S. M. El-Kholy conceived and designed the experiments, prepared figures and/or tables, authored or reviewed drafts of the article, and approved the final draft.
- Diaa Abd El-Moneim conceived and designed the experiments, performed the experiments, analyzed the data, prepared figures and/or tables, authored or reviewed drafts of the article, and approved the final draft.
- Ahmed M. Saad conceived and designed the experiments, performed the experiments, analyzed the data, authored or reviewed drafts of the article, and approved the final draft.

## Data Availability

The raw data is available in the Supplemental File.

## Supplemental Information

Supplemental information for this article can be found online at http://dx.doi.org/10.7717/peerj.19802#supplemental-information.

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
