# Peer review of "Impact of foliar application using amino acids, yeast extract, and algae extract in different concentrations on growth parameters, yield traits, grain quality, and nitrogen-related parameters of wheat in arid environments"

_PeerJ, doi:10.7717/peerj.19802_

## Round 0.1 · original submission · Major Revisions

Your manuscript requires major revisions.

Reviewer 1 ·

Basic reporting

- The manuscript is generally well written, with a clear structure and logical flow. The introduction provides adequate background and justifies the research question effectively.
- The literature review is comprehensive and up to date, referencing relevant studies from reputable journals.
- Tables and figures are appropriate and informative. However, the readability of Figures 2 (PCA biplot) and 3 (heatmap) could be improved by increasing label clarity and contrast.
- There are some repetitive phrases, especially in the Abstract and Discussion (e.g., “enhanced growth and yield”), which could be replaced with more varied academic vocabulary to improve readability.
- Minor typographical issues exist, including the repeated use of "Algea" instead of the correct spelling, “Algae,” which should be corrected throughout the manuscript and figures.

Experimental design

- The study design is robust, with a randomized complete block design (RCBD) applied over two growing seasons, ensuring replicability and reliability of results.
- Treatment applications are well defined. The foliar treatments and their chemical compositions are sufficiently described, although the complete composition of the yeast and algae extracts should be reported more clearly if available.
- Methodological details, including plot size, application timing, sampling procedure, and analytical techniques, are appropriately described, enabling replication.
- The statistical analysis includes ANOVA, LSD tests, and multivariate techniques such as PCA and heatmap clustering, which are well suited for this type of data.

Validity of the findings

- The results are presented in a detailed and structured manner, and the data interpretation aligns with the reported findings.
- Repeated trends across both seasons add credibility to the consistency and robustness of the outcomes.
- Amino acids at 3 ml/L consistently outperformed the other treatments in all growth and nitrogen-use metrics, and the superiority of higher concentrations is statistically supported.
- Nonetheless, the discussion would benefit from briefly addressing the economic feasibility of these treatments in real-world applications, as this would strengthen the practical implications of the findings.

Additional comments

- Language and Style: Revise long and repetitive sentences for conciseness and stylistic variety. Consider seeking a professional language edit to refine expression.
- Improve clarity of PCA biplot and heatmap (Figures 2 and 3). Increase font size and contrast, and include clear legends.
- Standardize terminology throughout the manuscript. Correct "Algea" to "Algae" in all instances.
- Add a short paragraph in the discussion regarding the cost-effectiveness of the foliar treatments.

Reviewer 2 ·

Basic reporting

The narrative of the article is written in a very understandable language.
The literature is quite up to date.
Data in tables and graphs are clearly displayed.
But the spelling errors I suggested in the attached file should be corrected.

Experimental design

Research question well defined, relevant & meaningful.

Validity of the findings

The conclusion part contains too many repetitive expressions.
The narrative should be improved in order not to bore the reader.
Unfortunately, the narration is very tiring for the reader.
Conclusions are well stated, linked to original research question & limited to supporting results.

Additional comments

Especially the results section should be rewritten more fluently. There are too many repetitive expressions.

Annotated reviews are not available for download in order to protect the identity of reviewers who chose to remain anonymous.

Reviewer 3 ·

Basic reporting

Thanks for considering me to review the manuscript entitled: "Impact of foliar nutrition using amino acids, yeast extract, and algae extract in different concentrations on growth parameters, yield traits, grain quality, and nitrogen-related parameters of wheat in arid environments".

The manuscript addresses a highly relevant topic in sustainable agriculture, focusing on the use of bio-stimulants to enhance wheat productivity and nitrogen efficiency in arid environments.

The manuscript provides valuable insights into the comparative effects of foliar-applied amino acids, yeast extract, and algae extract on wheat under arid conditions.

Experimental design

Including multiple concentrations and using multivariate analyses (PCA, heatmap) strengthens scientific contribution.

The experimental design is robust, the data are comprehensive, and the findings are well-supported by statistical analyses.

The section of Materials and Methods is well-structured and well-described.

Ensure all tables and figures are clearly labelled, formatted, and referenced in the correct order. The PCA and heatmap analyses are valuable but require clearer interpretation for readers less familiar with multivariate statistics. Briefly explain what the principal components represent and how to interpret the heatmap colour scale.

Validity of the findings

The manuscript is well-structured, with clear objectives, detailed methodology, and thorough discussion.

The manuscript has strong scientific merit but requires major revisions for clarity and completeness.

Addressing the following points will significantly strengthen the manuscript and enhance its impact.

Additional comments

The manuscript is generally well-written, but there are some grammatical errors. A thorough language edit is recommended. Use consistent terminology throughout (e.g., "bio-stimulant" vs. "biostimulant"; "foliar nutrition" vs. "foliar application").

The abstract is comprehensive; please clearly state the main conclusion in the final sentence.
The introduction could benefit from a brief overview of previous studies on foliar bio-stimulants in cereals, highlighting gaps addressed by this study.

The discussion effectively relates findings to previous studies and provides plausible physiological explanations for the observed effects. Please focus more on mechanisms, implications for sustainable agriculture, and potential limitations.

The conclusions are supported by the data, but could be more concise. Emphasise the practical recommendations for farmers and policymakers.

Ensure all references are up-to-date and formatted according to journal guidelines. Double-check for consistency between in-text citations and the reference list.

---

## Round 0.2 · accepted · Accept

I accept your manuscript after the changes you have made.

Reviewer 1 ·

Basic reporting

The revised manuscript is much clearer, more consistent, and better written. The authors have decisively addressed earlier concerns regarding repetitive and cumbersome phrasing in the Results and Conclusions sections. These parts are now more concise and engaging, which enhances readability. Spelling inconsistencies, such as the use of "Algea" instead of "Algae" have been thoroughly corrected throughout the manuscript and in all figures. The terminology is now consistent, with 'foliar application' and 'biostimulant' used uniformly. The overall narrative flows logically and clearly. The figures and tables are presented clearly and are informative.

Experimental design

The original study design was robust, and no issues were raised in the previous version concerning its structure. The authors had described the methodology in sufficient detail, allowing for reproducibility. This revision maintains the soundness of all methodological elements, with minor enhancements in clarity applied where necessary. The manuscript unquestionably meets PeerJ's standards for rigor and transparency in experimental design.

Validity of the findings

The findings are solidly supported by data and statistically analysed using appropriate methods (ANOVA, LSD, PCA, heatmap). In my initial review, I highlighted issues with readability due to repeated phrasing, particularly in the Results section. These have now been corrected, and the final version presents the data in a fluent and scientifically appropriate manner. The conclusions have been revised to align more closely with the study's results, and now clearly outline practical implications, especially for sustainable agriculture in arid environments. I find the interpretation valid and the scientific conclusions justified by the evidence.

Reviewer 2 ·

Basic reporting

The authors
have made all necessary corrections, taking into account the previously mentioned suggestions. The abstract is written in sufficient length and content.
Although the introduction is a bit long, the rich content it contains has reinforced the subject very well. Since the added sources are quite up-to-date, they have enriched the article.

Experimental design

There is no problem in the material and method section.

Validity of the findings

The discussion and conclusion sections have been thoroughly corrected in line with the suggestions.
Conclusions are well stated, linked to original research question & limited to supporting results.

Additional comments

I added a small correction to the introduction. Other than that, there doesn't seem to be any problem. Thanks.

Annotated reviews are not available for download in order to protect the identity of reviewers who chose to remain anonymous.